# Unsupervised Domain Adaptation with Dynamics-Aware Rewards in Reinforcement Learning

**Jinxin Liu**[124]   **Hao Shen**[3*] **Donglin Wang**[24†]   **Yachen Kang**[124]   **Qiangxing Tian**[124]
[1] Zhejiang University.   [2] Westlake University.   [3] UC Berkeley.
[4] Institute of Advanced Technology, Westlake Institute for Advanced Study.
liujinxin@westlake.edu.cn, haoshen@berkeley.edu,
{wangdonglin, kangyachen, tianqiangxing}@westlake.edu.cn

## Abstract

Unsupervised reinforcement learning aims to acquire skills without prior goal representations, where an agent automatically explores an open-ended environment to represent goals and learn the goal-conditioned policy. However, this procedure is often time-consuming, limiting the rollout in some potentially expensive target environments. The intuitive approach of training in another interaction-rich environment disrupts the reproducibility of trained skills in the target environment due to the dynamics shifts and thus inhibits direct transferring. Assuming free access to a source environment, we propose an unsupervised domain adaptation method to identify and acquire skills across dynamics. Particularly, we introduce a KL regularized objective to encourage emergence of skills, rewarding the agent for both discovering skills and aligning its behaviors respecting dynamics shifts. This suggests that both dynamics (source and target) shape the reward to facilitate the learning of adaptive skills. We also conduct empirical experiments to demonstrate that our method can effectively learn skills that can be smoothly deployed in target.

## 1   Introduction

Recently, the machine learning community has devoted attention to unsupervised reinforcement learning (RL) to acquire useful skills, ie, the problem of automatic discovery of a goal-conditioned policy and its corresponding goal space [8]. As shown in Figure 1 (left), the standard training procedure of learning skills in an unsupervised way follows: (1) representing goals, consisting of automatically generating the goal distribution $p(g)$ and the corresponding goal-achievement reward function $r_g$; (2) learning the goal-conditioned policy $\pi_\theta$ with the acquired $p(g)$ and $r_g$. Leveraging fully autonomous interaction with the environment, the agent sets up goals, builds the goal-achievement reward function, and extrapolates the goal-conditioned policy in parallel by adopting off-the-shelf RL methods [40, 19]. While we can obtain skills without any prior goal representations ($p(g)$ and $r_g$) in an unsupervised way, a major drawback of this approach is that it requires a large amount of rollout steps to represent goals and learn the policy itself, together. This procedure is often impractical in some target environments (eg, the robot in real world), where online interactions are time-consuming and potentially expensive.

That said, there often exist environments that resemble in structure (dynamics) yet provide more accessible rollouts (eg, unlimited in simulators). For problems with such source environments available, training the policy in a source environment significantly reduces the cost associated with interaction in the target environment. Critically, we can train a policy in one environment and deploy it in another by utilizing their structural similarity and the excess of interaction. Considering the navigation in a room, we can learn arbitrary skills through the active exploration in a source simulated

---

*Work was done at Westlake University. † Corresponding author.

35th Conference on Neural Information Processing Systems (NeurIPS 2021).

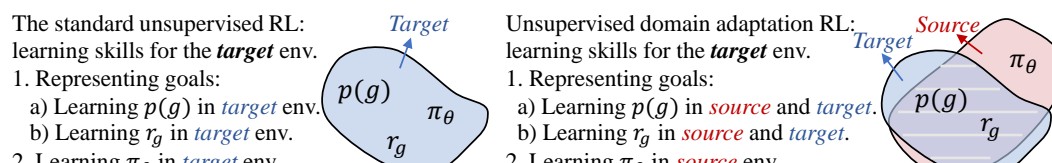

The standard unsupervised RL: learning skills for the **target** env.
1. Representing goals:
   a) Learning $p(g)$ in *target* env.
   b) Learning $r_g$ in *target* env.
2. Learning $\pi_\theta$ in *target* env.

*Target*

$p(g)$  $\pi_\theta$

$r_g$

Unsupervised domain adaptation RL: learning skills for the **target** env.
1. Representing goals:
   a) Learning $p(g)$ in *source* and *target*.
   b) Learning $r_g$ in *source* and *target*.
2. Learning $\pi_\theta$ in *source* env.

*Source*  *Target*  $\pi_\theta$

$p(g)$  $r_g$

Figure 1: The training procedures of (left) the standard unsupervised RL in a single target environment, and (right) the unsupervised domain adaptation in RL with a pair of source and target environments. $p(g)$: the goal distribution; $r_g$: the goal-achievement reward function; $\pi_\theta$: the goal-conditioned policy.

room (with different layout or friction) before the deployment in the target room. However, it is reasonable to suspect that the learned skills overfit the training environment, the dynamics of which, dictating the goal distribution and reward function, implicitly shape goal representation and guide policy acquisition. Such deployment would then make learned skills struggle to adapt to new, unseen environments and produce a large drop in performance in target due to the dynamics shifts, as shown in Figure 2 (top). In this paper, we overcome the limitations (of limited rollout in target and dynamics shifts) associated with the *(source, target)* environments pair through unsupervised domain adaptation.

In practice, while performing a *full* unsupervised RL method in target that represents goals and captures all of them for learning the entire goal-conditioned policy (Figure 1 left) can be extremely challenging with the limited rollout steps, learning a model for *only* (partially) representing goals is much easier. This gives rise to learning the policy in source and taking the limited rollouts in target into account only for identifying the goal representations, which further shape the policy. As shown in Figure 1 (right), we represent goals in both environments while optimizing the policy only in the source environment, alleviating the excessive need for rollout steps in the target environment.

Furthermore, we introduce a KL regularization to address the challenge of dynamics shifts. This objective allows us to incorporate a reward modification into the goal-achievement reward function in the standard unsupervised RL, aligning the trajectory induced in the target environment against that induced in the source by the same policy. Importantly, it enables useful inductive biases towards the target dynamics: it allows the agent to specifically pursue skills that are competent in the target dynamics, and penalizes the agent for exploration in the source where the dynamics significantly differ. As shown in Figure 2 (bottom), the difference in dynamics (a wall in the target while no wall in the source) will pose a penalty when the agent attempts to go through an area in the source wherein the target stands a wall. Thus, skills learned in source with such modification are adaptive to the target.

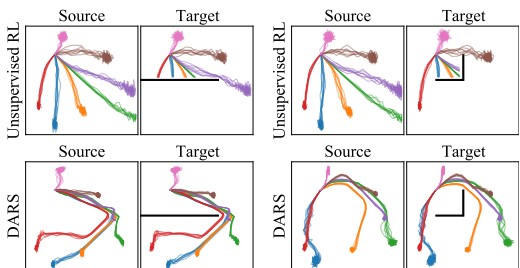

Figure 2: Skills *learned in the source environment*, each represented by a distinct color, are deployed in the source and target respectively. *Top plots* depict states visited by the standard unsupervised RL method, where skills fail to run in the target environment. *Bottom plots* depict trajectories induced by policy $\pi_\theta$ trained with our DARS, resulting in successful deployment in the target environment.

We name our method unsupervised domain adaptation with dynamics-aware rewards (DARS), suggesting that source and target dynamics both shape $r_g$: (1) we employ a latent-conditioned probing policy in the source to represent goals [31], making the goal-achievement reward source-oriented, and (2) we adopt two classifiers [11] to provide reward modification derived from the KL regularization. This means that the repertoires of skills are well shaped by the dynamics of both the source and target. Formally, we further analyze the conditions under which our DARS produces a near-optimal goal-conditioned policy for the target environment. Empirically, we demonstrate that our objective can obtain dynamics-aware rewards, enabling the goal-conditioned policy learned in a source to perform well in the target environment in various settings (stable and unstable settings, and sim2real).

## 2 Preliminaries

**Multi-goal Reinforcement Learning:** We formalize the multi-goal reinforcement learning (RL) as a goal-conditioned Markov Decision Process (MDP) defined by the tuple $\mathcal{M}_\mathcal{G} = \{S, A, \mathcal{P}, \mathcal{R}_G, \gamma, \rho_0\}$,

where $S$ denotes the state space and $A$ denotes the action space. $\mathcal{P} : S \times A \times S \to \mathbb{R}_{\geq 0}$ is the transition probability density. $\mathcal{R}_G \triangleq \{G, r_g, p(g)\}$, where $G$ denotes the space of goals, $r_g$ denotes the corresponding goal-achievement reward function $r_g : G \times S \times A \times S \to \mathbb{R}$, and $p(g)$ denotes the given goal distribution. $\gamma$ is the discount factor and $\rho_0$ is the initial state distribution. Given a $g \sim p(g)$, the $\gamma$-discounted return $R(g, \tau)$ of a goal-oriented trajectory $\tau = (s_0, a_0, s_1, \ldots, s_T)$ is $\Sigma_{t=0}^{T-1} \gamma^t r_g(s_t, a_t, s_{t+1})$. Building on the universal value function approximators (UVFA, Schaul et al. [38]), the standard multi-goal RL seeks to learn a unique goal-conditioned policy $\pi_\theta : A \times S \times G \to \mathbb{R}$ to maximize the objective $\mathbb{E}_{\mathcal{P}, \rho_0, \pi_\theta, p(g)}[R(g, \tau)]$, where $\theta$ denotes the parameter of the policy.

**Unsupervised Reinforcement Learning:** In unsupervised RL, the agent is set in an open-ended environment without any pre-defined goals or related reward functions. The agent aims to acquire a repertoire of skills. Following Colas et al. [8], we define skills as the association of goals and the goal-conditioned policy to reach them. The unsupervised skill acquisition problem can now be modeled by a goal-free MDP $\mathcal{M} = \{S, A, \mathcal{P}, \gamma, \rho_0\}$ that only characterizes the agent, its environment and their possible interactions. As shown in Figure 1 (left), the agent needs to autonomously interact with the environment and (1) *learn goal representations* (eg, discovering the goal distribution $p(g)$ and learning the corresponding reward $r_g$), and (2) *learn the goal-conditioned policy* $\pi_\theta$ as in multi-goal RL.

Here we define a universal (information theoretic) objective for learning the goal-conditioned policy $\pi_\theta$ in unsupervised RL, maximizing the mutual information $\mathcal{I}_{\mathcal{P}, \rho_0, \pi_\theta}(g; \tau)$ between the goal $g$ and the trajectory $\tau$ induced by policy $\pi_\theta$ running in the environment $\mathcal{M}$ (with $\mathcal{P}$ and $\rho_0$),

$$\max \, \mathcal{I}_{\mathcal{P}, \rho_0, \pi_\theta}(g; \tau) = \mathcal{H}(g) - \mathcal{H}(g|\tau) = \mathcal{H}(g) + \mathbb{E}_{\mathcal{P}, \rho_0, \pi_\theta, p(g)}[\log p(g|\tau)]. \tag{1}$$

For representing goals, the specific manifold of the goal space could be a set of *latent variables* (eg, one-hot indicators) or *perceptually-specific goals* (eg, the joint torques of ant). In the absence of any prior knowledge about $p(g)$, the maximum of $\mathcal{H}(g)$ will be achieved by fixing the distribution $p(g)$ to be uniform over all $g \in G$. The second term $\mathbb{E}_{\mathcal{P}, \rho_0, \pi_\theta, p(g)}[\log p(g|\tau)]$ in Equation 1 is analogous to the objective in the standard multi-goal RL, where the return $R(g, \tau)$ can be seen as the embodiment of $\log p(g|\tau)$. The objective specifically for learning $r_g$ in $p(g|\tau)$ is normally optimized by lens of the generative loss [33] or the contrastive loss [42]. With the learned goal distribution $p(g)$ and reward $r_g$, it is straightforward to learn the goal-conditioned policy $\pi_\theta$ using standard RL algorithms [40, 19]. In general, optimizations iteratively alternate for representing goals (including both goal-distribution $p(g)$ and reward function $r_g$) and learning the goal-conditioned policy $\pi_\theta$, as shown in Figure 1 (left).

## 3 Unsupervised Domain Adaptation with Dynamics-Aware Rewards

### 3.1 Problem Formulation

Our work addresses domain adaptation in unsupervised RL, raising expectations that an agent trained without prior goal representations ($p(g)$ and $r_g$) in one environment can perform purposeful tasks in another. Following Wulfmeier et al. [54], we also focus on the domain adaptation of the dynamics, as opposed to states. In this work, we consider two environments characterized by MDPs $\mathcal{M}_\mathcal{S}$ (the source environment) and $\mathcal{M}_\mathcal{T}$ (the target environment), the dynamics of which are $\mathcal{P}_\mathcal{S}$ and $\mathcal{P}_\mathcal{T}$ respectively. Both MDPs share the same state and action spaces $S, A$, discount factor $\gamma$ and initial state distribution $\rho_0$, while differing in the transition distributions $\mathcal{P}_\mathcal{S}, \mathcal{P}_\mathcal{T}$. Since the agent does not directly receive $\mathcal{R}_\mathcal{G}$ from either environment, we adopt the information theoretic $\mathcal{I}_{\mathcal{P}, \rho_0, \pi_\theta}(g; \tau)$ to acquire skills, equivalently learning a goal-conditioned policy $\pi_\theta$ that achieves distinguishable trajectory by maximizing this objective. For brevity, we now omit the $\rho_0$ term discussed in Section 2.

In our setup, agents can freely interact with the source $\mathcal{M}_\mathcal{S}$. However, it has limited access to rollouts in the target $\mathcal{M}_\mathcal{T}$ with which are insufficient to train a policy. To ensure that all potential trajectories in the target $\mathcal{M}_\mathcal{T}$ can be attempted in the source environment, we make the following assumption:

**Assumption 1.** *There is no transition that is possible in the target environment $\mathcal{M}_\mathcal{T}$ but impossible in the source environment $\mathcal{M}_\mathcal{S}$: $\mathcal{P}_\mathcal{T}(s_{t+1}|s_t, a_t) > 0 \implies \mathcal{P}_\mathcal{S}(s_{t+1}|s_t, a_t) > 0$.*

### 3.2 Domain Adaptation in Unsupervised RL

We aim to acquire skills trained in the source environment $\mathcal{M}_\mathcal{S}$, which can be deployed in the target environment $\mathcal{M}_\mathcal{T}$. To facilitate the unsupervised learning of skills for the target environment $\mathcal{M}_\mathcal{T}$

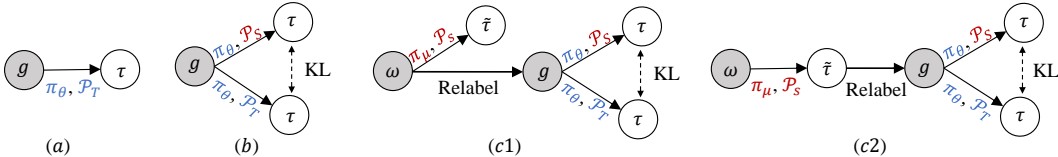

Figure 3: Graphical models of *(a)* the standard unsupervised RL, and DARS with goals *(b)* directly inputted, *(c1)* relabeled with latent variable $\omega$, and *(c2)* relabeled with state induced by probing policy.

(with transition dynamics $\mathcal{P}_\mathcal{T}$), we maximize the mutual information between the goal $g$ and the trajectory $\tau$ induced by the goal-conditioned policy $\pi_\theta$ over dynamics $\mathcal{P}_\mathcal{T}$, as shown in Figure 3 (a):

$$\mathcal{I}_{\mathcal{P}_\mathcal{T},\pi_\theta}(g;\tau). \tag{2}$$

However, since interaction with the target environment $\mathcal{M}_\mathcal{T}$ is restricted, acquiring the goal-conditioned policy $\pi_\theta$ by optimizing the mutual information above is intractable. We instead maximize the mutual information in the source environment $\mathcal{I}_{\mathcal{P}_\mathcal{S},\pi_\theta}(g;\tau)$ modified by a KL divergence of trajectories induced by the goal-conditioned policy $\pi_\theta$ in both environments (Figure 3 b):

$$\mathcal{I}_{\mathcal{P}_\mathcal{S},\pi_\theta}(g;\tau) - \beta D_{\mathrm{KL}}\left(p_{\mathcal{P}_\mathcal{S},\pi_\theta}(g,\tau)\|p_{\mathcal{P}_\mathcal{T},\pi_\theta}(g,\tau)\right), \tag{3}$$

where $\beta > 0$ is the regularization coefficient, $p_{\mathcal{P}_\mathcal{S},\pi_\theta}(g,\tau)$ and $p_{\mathcal{P}_\mathcal{T},\pi_\theta}(g,\tau)$ denote the joint distributions of the goal $g$ and the trajectory $\tau$ induced by policy $\pi_\theta$ in source $\mathcal{M}_\mathcal{S}$ and target $\mathcal{M}_\mathcal{T}$ respectively.

Intuitively, maximizing the mutual information term rewards distinguishable pairs of trajectories and goals, while minimizing the KL divergence term penalizes producing a trajectory that cannot be followed in the target environment. In other words, the KL term aligns the probability distributions of the mutual-information-maximizing trajectories under the two environment dynamics $\mathcal{P}_\mathcal{S}$ and $\mathcal{P}_\mathcal{T}$. This indicates that the dynamics of both environments ($\mathcal{P}_\mathcal{S}$ and $\mathcal{P}_\mathcal{T}$) shape the goal-conditioned policy $\pi_\theta$ (even though trained in the source $\mathcal{P}_\mathcal{S}$), allowing $\pi_\theta$ to adapt to the shifts in dynamics.

Building on the KL regularized objective in Equation 3, we introduce how to effectively represent goals: generating the goal distribution and acquiring the (partial) reward function. Here we assume the difference between environments in their dynamics negligibly affects the goal distribution[2]. Therefore, we follow GPIM [31] and train a latent-conditioned probing policy $\pi_\mu$. The probing policy $\pi_\mu$ explores the source environment and represents goals for the source to train the goal-conditioned policy $\pi_\theta$ with. Specifically, the probing policy $\pi_\mu$ is conditioned on a latent variable $\omega \sim p(\omega)$[3] and aims to generate diverse trajectories that are further relabeled as goals for the goal-conditioned $\pi_\theta$. Such goals can take the form of the latent variable $\omega$ itself (Figure 3 c1) or the final state of a trajectory (Figure 3 c2). We jointly optimize the previous objective in Equation 3 with the mutual information between $\omega$ and the trajectory $\tilde{\tau}$ induced by $\pi_\mu$ in source, and arrive at the following overall objective:

$$\max \mathcal{J}(\mu,\theta) \triangleq \mathcal{I}_{\mathcal{P}_\mathcal{S},\pi_\mu}(\omega;\tilde{\tau}) + \mathcal{I}_{\mathcal{P}_\mathcal{S},\pi_\theta}(g;\tau) - \beta D_{\mathrm{KL}}\left(p_{\mathcal{P}_\mathcal{S},\pi_\theta}(g,\tau)\|p_{\mathcal{P}_\mathcal{T},\pi_\theta}(g,\tau)\right), \tag{4}$$

where the context between $p(g)$ and $p(\omega)$ are specified by the graphic model in Figure 3 (c1 or c2). Note that this objective (Equation 4) explicitly decouples the goal representing (with $\pi_\mu$) and the policy learning (wrt $\pi_\theta$), providing a foundation for the theoretical guarantee in Section 3.4.

### 3.3 Optimization with Dynamics-Aware Rewards

Similar to Goyal et al. [16], we take advantage of the data processing inequality (DPI [3]) which implies $\mathcal{I}_{\mathcal{P}_\mathcal{S},\pi_\theta}(g;\tau) \geq \mathcal{I}_{\mathcal{P}_\mathcal{S},\pi_\theta}(\omega;\tau)$ from the graphical models in Figure 3 (c1, c2). Consequently, maximizing $\mathcal{I}_{\mathcal{P}_\mathcal{S},\pi_\theta}(g;\tau)$ can be achieved by maximizing the information of $\omega$ encoded progressively to $\pi_\theta$. We therefore obtain the lower bound of Equation 4:

$$\mathcal{J}(\mu,\theta) \geq \mathcal{I}_{\mathcal{P}_\mathcal{S},\pi_\mu}(\omega;\tilde{\tau}) + \mathcal{I}_{\mathcal{P}_\mathcal{S},\pi_\theta}(\omega;\tau) - \beta D_{\mathrm{KL}}\left(p_{\mathcal{P}_\mathcal{S},\pi_\theta}(g,\tau)\|p_{\mathcal{P}_\mathcal{T},\pi_\theta}(g,\tau)\right). \tag{5}$$

---

[2]See Appendix D for the extension when $\mathcal{M}_\mathcal{S}$ and $\mathcal{M}_\mathcal{T}$ have different goal distributions.

[3]Following DIAYN [10] and DADS [43], we set $p(\omega)$ as a fixed prior.

For the first term $\mathcal{I}_{\mathcal{P}_S,\pi_\mu}(\omega;\tilde{\tau})$ and the second term $\mathcal{I}_{\mathcal{P}_S,\pi_\theta}(\omega;\tau)$, we derive the state-conditioned Markovian rewards following Jabri et al. [24]:

$$\mathcal{I}_{\mathcal{P},\pi}(\omega;\tau) \geq \frac{1}{T}\sum_{t=0}^{T-1}\left(\mathcal{H}(\omega) - \mathcal{H}(\omega|s_{t+1})\right) = \mathcal{H}(\omega) + \mathbb{E}_{p_{\mathcal{P},\pi}(\omega,s_{t+1})}\left[\log p(\omega|s_{t+1})\right] \quad (6)$$

$$\geq \mathcal{H}(\omega) + \mathbb{E}_{p_{\mathcal{P},\pi}(\omega,s_{t+1})}\left[\log q_\phi(\omega|s_{t+1})\right], \quad (7)$$

where $p_{\mathcal{P},\pi}(\omega,s_{t+1}) = p(\omega)p_{\mathcal{P},\pi}(s_{t+1}|\omega)$, and $p_{\mathcal{P},\pi}(s_{t+1}|\omega)$ refers to the state distribution (at time step $t+1$) induced by policy $\pi$ conditioned on $\omega$ under the environment dynamics $\mathcal{P}$; the lower bound in Equation 7 derives from training a discriminator network $q_\phi$ due to the non-negativity of KL divergence, $\mathbb{E}_{p_\pi(s_{t+1})}\left[D_{\mathrm{KL}}(p(\omega|s_{t+1})||q_\phi(\omega|s_{t+1}))\right] \geq 0$. Intuitively, the new bound rewards the discriminator $q_\phi$ for summarizing agent's behavior with $\omega$ as well as encouraging a variety of states.

With the bound above, we construct the lower bound of the mutual information terms in Equation 5, taking the same discriminator $q_\phi$:

$$\mathcal{F}_\mathcal{I} \triangleq \mathcal{I}_{\mathcal{P}_S,\pi_\mu}(\omega;\tilde{\tau}) + \mathcal{I}_{\mathcal{P}_S,\pi_\theta}(\omega;\tau) \geq 2\mathcal{H}(\omega) + \mathbb{E}_{p_{\mathrm{joint}}}\left[\log q_\phi(\omega|\tilde{s}_{t+1}) + \log q_\phi(\omega|s_{t+1})\right], \quad (8)$$

where $p_{\mathrm{joint}}$ denotes the joint distribution of $\omega$, states $\tilde{s}_{t+1}$ and $s_{t+1}$. The states $\tilde{s}_{t+1}$ and $s_{t+1}$ are induced by the probing policy $\pi_\mu$ conditioned on the latent variable $\omega$ and the policy $\pi_\theta$ conditioned on the relabeled goals respectively, both in the source environment (Figure 3 c1, c2).

Now, we are ready to characterize the KL term in Equation 5. Note that only the transition probabilities terms ($\mathcal{P}_S$ and $\mathcal{P}_T$) differ since agent follows the same policy $\pi_\theta$ in the two environments. This conveniently leads to the expansion of the KL divergence term as a sum of differences in log likelihoods of the transition dynamics: expansion $p_{\mathcal{P},\pi_\theta}(g,\tau) = p(g)\rho_0(s_0)\prod_{t=0}^{T-1}[\mathcal{P}(s_{t+1}|s_t,a_t)\pi_\theta(a_t|s_t,g)]$, where $\mathcal{P} \in \{\mathcal{P}_S,\mathcal{P}_T\}$, gives rise to the following simplification of the KL term in Equation 5:

$$\beta D_{\mathrm{KL}}\left(p_{\mathcal{P}_S,\pi_\theta}(g,\tau)||p_{\mathcal{P}_T,\pi_\theta}(g,\tau)\right) = \mathbb{E}_{\mathcal{P}_S,\pi_\theta}\left[\beta\Delta r(s_t,a_t,s_{t+1})\right], \quad (9)$$

where the reward modification $\Delta r(s_t,a_t,s_{t+1}) \triangleq \log\mathcal{P}_S(s_{t+1}|s_t,a_t) - \log\mathcal{P}_T(s_{t+1}|s_t,a_t)$.

Combining the lower bound of the mutual information terms (Equation 8) and the KL divergence term pursuing the aligned trajectories in two environments (Equation 9), we optimize $\mathcal{J}(\mu,\theta)$ by maximizing the following lower bound:

$$2\mathcal{H}(\omega) + \mathbb{E}_{p_{\mathrm{joint}}}\left[\log q_\phi(\omega|\tilde{s}_{t+1}) + \log q_\phi(\omega|s_{t+1})\right]$$
$$- \mathbb{E}_{\mathcal{P}_S,\pi_\theta}\left[\beta\Delta r(s_t,a_t,s_{t+1})\right]. \quad (10)$$

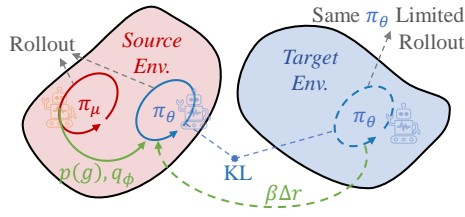

Associted reward for $\pi_\theta$ : $r_g = \log q_\phi - \beta\Delta r$

Figure 4: Framework of DARS: the latent-conditioned probing policy $\pi_\mu$ provides $p(g)$ and $q_\phi$ for learning goal-conditioned $\pi_\theta$, associated with the reward modification $\beta\Delta r$.

Overall, as shown in Figure 4, DARS rewards the goal-conditioned policy $\pi_\theta$ with the dynamics-aware rewards (associating $\log q_\phi$ with $\beta\Delta r$), where (1) $\log q_\phi$ is shaped by the source dynamics $\mathcal{P}_S$, and (2) $\beta\Delta r$ is derived from the difference of the two dynamics ($\mathcal{P}_S$ and $\mathcal{P}_T$). This indicates that the learned goal-conditioned policy $\pi_\theta$ is shaped by both source and target environments, holding the promise of acquiring adaptive skills for the target environment by training mostly in the source environment.

### 3.4 Optimality Analysis

Here we discuss the condition under which our method produces near-optimal skills for the target environment. We first mildly require that the most suitable policy for the target environment $\mathcal{M}_T$ does not produce drastically different trajectories in the source environment $\mathcal{M}_S$:

**Assumption 2.** Let $\pi^* = \arg\max_\pi \mathcal{I}_{\mathcal{P}_T,\pi}(g;\tau)$ be the policy that maximizes the (non-kl-regularized) objective in the target environment (Equation 2). Then the joint distributions of the goal and its trajectories differ in both environments by no more than a small number $\epsilon/\beta > 0$:

$$D_{\mathrm{KL}}\left(p_{\mathcal{P}_S,\pi^*}(g,\tau)||p_{\mathcal{P}_T,\pi^*}(g,\tau)\right) \leq \frac{\epsilon}{\beta}. \quad (11)$$

**Algorithm 1** DARS

1: **Input:** source and target MDPs $\mathcal{M}_\mathcal{S}$ and $\mathcal{M}_\mathcal{T}$; ratio $R$ of experience from source vs. target.
2: **Output:** goal-reaching policy $\pi_\theta$.
3: Initialize parameters $\mu$, $\theta$, $\phi$ and $\psi$.
4: Initialize buffers $\tilde{\mathcal{B}}_\mathcal{S}$, $\mathcal{B}_\mathcal{S}$ and $\mathcal{B}_\mathcal{T}$.
5: **for** $iter = 0, \dots, \text{MAX\_ITER}$ **do**
6:    Sample latent variable: $\omega \sim p(\omega)$.
7:    Collect probing data in source:
      $\tilde{\mathcal{B}}_\mathcal{S} \leftarrow \tilde{\mathcal{B}}_\mathcal{S} \cup \text{ROLLOUT}(\pi_\mu, \mathcal{M}_\mathcal{S}, \omega)$.
8:    Update discriminator $q_\phi$: $\phi \leftarrow \text{Update}(\phi, \tilde{\mathcal{B}}_\mathcal{S})$
9:    Set reward function for the probing policy $\pi_\mu$:
      $\tilde{r} = \log q_\phi(\omega|\tilde{s}_{t+1})$.
10:   Train probing policy $\pi_\mu$: $\mu \leftarrow \text{SAC}(\mu, \tilde{\mathcal{B}}_\mathcal{S}, \tilde{r})$.
11:   Relabel goals: # According to Figure 3 (c1, c2)
      $g \leftarrow \text{Relabel}(\omega, \tilde{\tau})$.
12:   Collect source data:
      $\mathcal{B}_\mathcal{S} \leftarrow \mathcal{B}_\mathcal{S} \cup \text{ROLLOUT}(\pi_\theta, \mathcal{M}_\mathcal{S}, g, \omega)$.
13:   **if** $iter \bmod R = 0$ **then**
14:       Collect target data:
          $\mathcal{B}_\mathcal{T} \leftarrow \mathcal{B}_\mathcal{T} \cup \text{ROLLOUT}(\pi_\theta, \mathcal{M}_\mathcal{T}, g)$.
15:   **end if**
16:   Update classifiers $q_\psi$ for computing $\Delta r$:
      $\psi \leftarrow \text{Update}(\psi, \mathcal{B}_\mathcal{S}, \mathcal{B}_\mathcal{T})$. (Equations 12, 13)
17:   Set reward function for $\pi_\theta$:
      $r_g \leftarrow \log q_\phi(\omega|s_{t+1}) - \beta \Delta r(s_t, a_t, s_{t+1})$.
18:   Train policy $\pi_\theta$: $\theta \leftarrow \text{SAC}(\theta, \mathcal{B}_\mathcal{S}, r_g)$.
19: **end for**

Given a desired joint distribution $p^*(g, \tau)$ (inferred from a potential goal representation), our problem can be reformulated as finding a closest match [29, 28]. Consequently, we quantify the optimality of a policy $\pi$ by measuring $D_{\text{KL}}\left(p_{\mathcal{P},\pi}(g,\tau)\|p_\mathcal{P}^*(g,\tau)\right)$, the discrepancy between its joint distribution and the desired one. With a potential goal representation, we prove that its joint distributions with the trajectories induced by our policy and the optimal one satisfy the following theoretical guarantee.

**Theorem 1.** *Let $\pi_{DARS}^*$ be the optimal policy that maximizes the KL regularized objective in the source environment (Equation 3), let $\pi^*$ be the policy that maximizes the (non-regularized) objective in the target environment (Equation 2), let $p_{\mathcal{P}_\mathcal{T}}^*(g,\tau)$ be the desired joint distribution of trajectory and goal in the target (with the potential goal representations), and assume that $\pi^*$ satisfies Assumption 2. Then the following holds:*

$$D_{KL}\left(p_{\mathcal{P}_\mathcal{T}, \pi_{DARS}^*}(g,\tau)\|p_{\mathcal{P}_\mathcal{T}}^*(g,\tau)\right) \leq D_{KL}\left(p_{\mathcal{P}_\mathcal{T}, \pi^*}(g,\tau)\|p_{\mathcal{P}_\mathcal{T}}^*(g,\tau)\right) + 2\sqrt{\frac{2\epsilon}{\beta}}L_{max},$$

*where $L_{max}$ refers to the worst case absolute difference between log likelihoods of the desired joint distribution and that induced by a policy.*

Please see Appendix C for more details and the proof of the theorem. Note that Theorem 1 requires a potential goal representation, which can be precisely provided by the probing policy $\pi_\mu$ in Equation 4.

### 3.5 Implementation

As shown in Algorithm 1, we alternately train the probing policy $\pi_\mu$ and the goal-conditioned policy $\pi_\theta$ by optimizing the objective in Equation 10 with respect to $\mu$, $\phi$, $\theta$ and $\Delta r$. In the first phase, we update $\pi_\mu$ with reward $\tilde{r} = \log q_\phi(\omega|\tilde{s}_{t+1})$. This is compatible with most RL methods and we refer to SAC here. We additionally optimize discriminator $q_\phi$ with SGD to maximizing $\mathbb{E}_{\omega, \tilde{s}_{t+1}}\left[q_\phi(\omega|\tilde{s}_{t+1})\right]$ at the same time. Similarly, $\pi_\theta$ is updated with $r_g = \log q_\phi(\omega|s_{t+1}) - \beta \Delta r$ by SAC in the second phase, where $\pi_\theta$ also collects (limited) data in the target environment to approximate $\Delta r$ by training two classifiers $q_\psi$ (wrt state-action $q_\psi^{sa}$ and state-action-state $q_\psi^{sas}$) as in [11] according to Bayes' rule:

$$\max \ \mathbb{E}_{\mathcal{B}_\mathcal{S}}\left[\log q_\psi^{sas}(\text{source}|s_t, a_t, s_{t+1})\right] + \mathbb{E}_{\mathcal{B}_\mathcal{T}}\left[\log q_\psi^{sas}(\text{target}|s_t, a_t, s_{t+1})\right], \quad (12)$$

$$\max \ \mathbb{E}_{\mathcal{B}_\mathcal{S}}\left[\log q_\psi^{sa}(\text{source}|s_t, a_t)\right] + \mathbb{E}_{\mathcal{B}_\mathcal{T}}\left[\log q_\psi^{sa}(\text{target}|s_t, a_t)\right]. \quad (13)$$

Then, we have $\Delta r(s_t, a_t, s_{t+1}) = \log \frac{q_\psi^{sas}(\text{source}|s_t, a_t, s_{t+1})}{q_\psi^{sas}(\text{target}|s_t, a_t, s_{t+1})} - \log \frac{q_\psi^{sa}(\text{source}|s_t, a_t)}{q_\psi^{sa}(\text{target}|s_t, a_t)}$.

### 3.6 Connections to Prior Work

**Unsupervised RL:** Two representative unsupervised RL approaches acquire (diverse) skills by maximizing empowerment [10, 43] or minimizing surprise [4]. Liu et al. [31] also employs a latent-conditioned policy to explore the environment and relabels goals along with the corresponding reward,

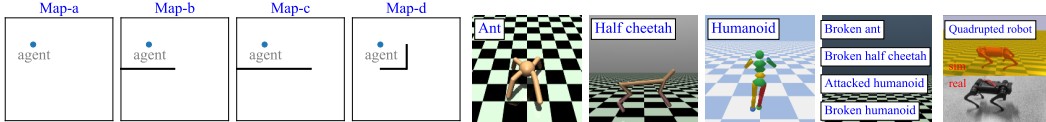

Figure 5: We evaluate our method in 10 *(source, target)* transition tasks, where the shifts in dynamics are either external (the map pairs and the attacked series) or internal (the broken series) to the robot.

which can be considered as a special case of DARS with identical source and target environments. However, none of these methods can produce skills tailored to new environments with dynamics shifts.

**Off-Dynamics RL:** Eysenbach et al. [11] proposes domain adaptation with rewards from classifiers (DARC), adopting the control as inference framework [29] to maximize $-D_{\mathrm{KL}}(p_{\mathcal{P}_{\mathcal{S}},\pi_\theta}(\tau)\|p^*_{\mathcal{P}_{\mathcal{T}}}(\tau))$, but this objective cannot be directly applied to the unsupervised setting. While we adopt the same classifier to provide the reward modification, one major distinction of our work is that we do not require a given goal distribution $p(g)$ or a prior reward function $r_g$. Moreover, assuming an extrinsic goal-reaching reward in the source environment (ie, the potential $p^*_{\mathcal{P}_{\mathcal{S}}}(\tau)$), our proposed DARS can be simplified to a *decoupled objective*: maximizing $-D_{\mathrm{KL}}(p_{\mathcal{P}_{\mathcal{S}},\pi_\theta}(\tau)\|p^*_{\mathcal{P}_{\mathcal{S}}}(\tau)) - \beta D_{\mathrm{KL}}(p_{\mathcal{P}_{\mathcal{S}},\pi_\theta}(\tau)\|p_{\mathcal{P}_{\mathcal{T}},\pi_\theta}(\tau))$. Particularly, DARC can be considered as a special case of our decoupled objective with the restriction — a prior goal specified by its corresponding reward and $\beta = 1$. In Appendix E, we show that the stronger pressure ($\beta > 1$) for the KL term to align the trajectories puts extra reward signals for the policy $\pi_\theta$ to be $\Delta r$ oriented while still being sufficient to acquire skills.

## 4 Related Work

The proposed DARS has interesting connections with unsupervised learning [10, 43] and transfer learning [55] in model-free RL. Adopting the self-supervised objective [26, 39, 2, 34], most approaches in this field consider learning features [18, 41] of high-dimensional (eg, image-based) states in the environment, then (1) adopt the non-parametric measurement function to acquire rewards [23, 33, 42, 53, 44, 32] or (2) enable policy transfer [23, 16, 17, 13, 22] over the learned features. These approaches can be seen as a procedure on the perception level [20], while we focus on the action level [20] wrt the transition dynamics of the environment, and we consider *both* cases (learning the goal-achievement reward function and enabling policy transfer between different environments).

Previous works on the action level [20] have either (1) focused on learning dynamics-oriented rewards in the unsupervised RL setting [21, 51, 49, 31] or (2) considered the transition-oriented modification in the supervised RL setting (given prior tasks described with reward functions or expert trajectories) [11, 54, 25, 14, 9, 52, 30]. Thus, the desirability of our approach is that the acquired reward function uncovers *both* the source dynamics ($q_\phi$) and the dynamics difference ($\beta \Delta r$) across source and target environment. Complementary to our work, several other works also encourage the emergence of a state-covering goal distribution [37, 6, 27] or enable transfer by introducing the regularization over policies [45, 15, 46, 47, 36, 48] instead of the adaptation over different dynamics.

## 5 Experiments

In this section, we aim to experimentally answer the following questions: (1) Can our method DARS learn diverse skills, in the source environment, that can be executed in the target environment and keep the same embodiment in the two environments? Specifically, can our proposed associated dynamics-aware rewards ($\log q_\phi - \beta \Delta r$) reveal the perceptible dynamics of the two environments? (2) Does DARS lead to better transferring in the presence of dynamics mismatch, compared to other related approaches, in both stable and unstable environments? (3) Can DARS contribute to acquiring behavioral skills under the sim2real circumstances, where the interaction in the real world is limited?

We adopt tuples *(source, target)* to denote the source and target environment pairs, with details of the corresponding MDPs in Appendix F.2. Illustrations of the environments are shown in Figure 5. For all tuples, we set $\beta = 10$ and the ratio of experience from the source environment vs. the target environment $R = 10$ (Line 13 in Algorithm 1). See Appendix F.3 for the other hyperparameters.

***Map.*** We consider the maze environments: *Map-a*, *Map-b*, *Map-c* and *Map-d*, where the wall can block the agent (a point), which can move around to explore the maze environment. For the domain adaptation tasks, we consider the following five *(source, target)* pairs: *(Map-a, Map-b)*, *(Map-a, Map-c)*, *(Map-a, Map-d)*, *(Map-b, Map-c)* and *(Map-b, Map-d)*.

***Mujoco.*** We use two simulated robots from OpenAI Gym [5]: half cheetah (*HC*) and ant. We define two new environments by crippling one of the joints of each robot (*B-HC* and *B-ant*) as described in [11], where *B-* is short for broken. The *(source, target)* pairs include: *(HC, B-HC)* and *(ant, B-ant)*.

***Humanoid.*** In this environment, a (source) simulated humanoid (*H*) agent must avoid falling in the face of the gravity disturbances. Two target environments each contain a humanoid *attacked* by blocks from a fixed direction (*A-H*) and a humaniod with a part of *broken* joints (*B-H*).

***Quadruped robot.*** We also consider the sim2real setting for transferring the simulated quadruped robot to a real quadruped robot. For more evident comparison, we break the left hind leg of the real-world robot (see Appendix F.2). We adopt *(sim-robot, real-robot)* to denote this sim2real transition.

## 5.1 Emergent Behaviors with DARS

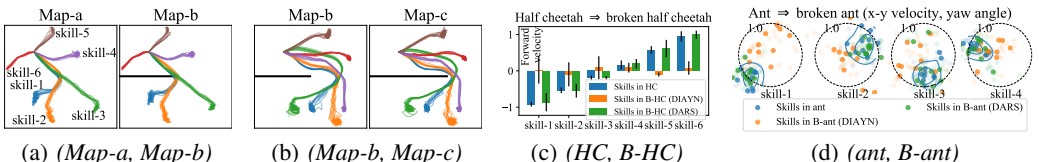

|     (a) *(Map-a, Map-b)*     |     (b) *(Map-b, Map-c)*     |     (c) *(HC, B-HC)*     |     (d) *(ant, B-ant)*     |

Figure 6: Visualization of skills. *(a, b)*: colored trajectories in *map* pairs depict the skills, learned with DARS, deployed in source (left) and target (right). *(c, d)*: colored bars and dots depict the velocity of each skill wrt different environments of *mujoco* and models. The variation (blue) across velocities for *HC* and *ant* corroborates the diversity of skills. DARS demonstrates its better adaptability by performing similar skills on broken agents (green) to the original ones (blue) while DIAYN (orange) fails.

**Visualization of the learned skills.** We first apply DARS to the *map* pairs and the *mujoco* pairs, where we learn the goal-conditioned policy $\pi_\theta$ in the source environments with our dynamics-aware rewards ($\log q_\phi - \beta \Delta r$). Here, we relabel the latent random variable $\omega$ as the goal $g$ for the goal-conditioned policy $\pi_\theta$: $g \triangleq \text{Relabel}(\pi_\mu, \omega, \tilde{\tau}) = \omega$ (Figure 3 c1). The learned skills are shown in Figures 2, 6 and Appendix E. We can see that the skills learned by our method keep the same embodiment when they are deployed in the source and target environments. If we directly apply the skills learned in the source environment (without $\beta \Delta r$), the dynamics mismatch is likely to disrupt the skills (see Figure 2 *top*, and the deployment of DIAYN in *half cheetah* and *ant* pairs in Figure 6).

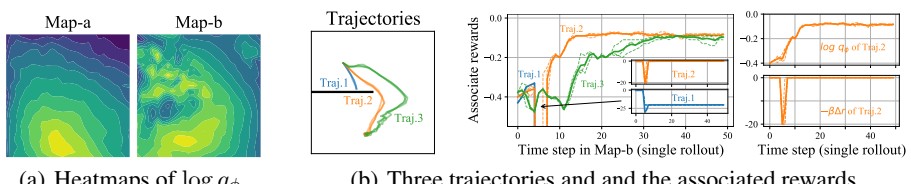

|     (a) Heatmaps of $\log q_\phi$.     |     (b) Three trajectories and and the associated rewards.     |

Figure 7: *(a)*: The value of $\log q_\phi$ in *Map-a* for *(Map-a, Map-b)* and $\log q_\phi$ in *Map-b* for *(Map-b, Map-c)*. *(b)*: Three trajectories in *Map-b* for the *(Map-b, Map-c)* task, and the recorded rewards.

**Visualizing the dynamics-aware rewards.** To gain more intuition that the proposed dynamics-aware rewards capture the perceptible dynamics of both the source and target environments and enable an adaptive policy for the target, we visualize the learned probing reward $\log q_\phi$ and the reward modification $\beta \Delta r$ throughout the training for *(Map-a, Map-c)* and *(Map-b, Map-c)* pairs in Figure 7.

The probing policy learns $q_\phi$ by summarizing the behaviors with the latent random variable $\omega$ in source environments. Setting *Map-a* as the source (Figure 7 (a) *left*), we can see that $\log q_\phi$ resembles the usual L2-norm-based punishment. Further, in the pair *(Map-b, Map-c)*, we can find that the learned $\log q_\phi$ is well shaped by the dynamics of the source environment *Map-b* (Figure 7 (a) *right*):

even if the agent simply moves in the direction of reward increase, it almost always sidesteps the wall and avoids the entrapment in a local optimal solution produced by the usual L2-norm based reward.

To see how the modification $\beta\Delta r$ guides the policy, we track three trajectories (with the same goal) and the associated rewards ($\log q_\phi - \beta\Delta r$) in the *(Map-b, Map-c)* task, as shown in Figure 7 (b). We see that *Traj.2* receives an incremental $\log q_\phi$ along the whole trajectory while a severe punishment from $\beta\Delta r$ around step 6. This indicates that *Traj.2* is inapplicable to the target dynamics (*Map-c*), even if it is feasible in the source (*Map-b*). With this modification, we indeed obtain the adaptive skills (eg. *Traj.3*) by training in the source. This answers our first question, where both dynamics (source and target) explicitly shape the associated rewards, guiding the skills to be domain adaptive.

## 5.2 Comparison with Baselines

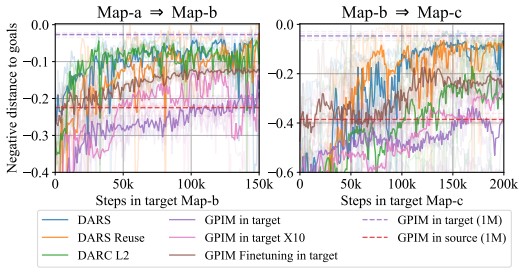

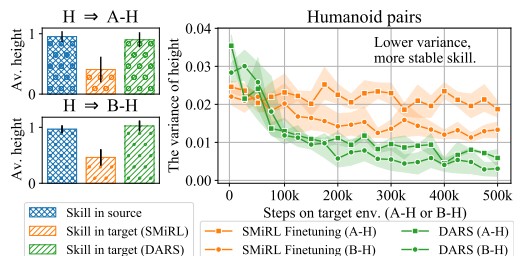

Figure 8: Comparison (training process) with alternative methods for learning skills for target environments. We plot each random seed as a transparent line; each solid line corresponds to the average across four random seeds; the dashed lines denote the performance of trained policies.

Figure 9: (left): The visualization of skills for humanoid (avoid falling) and the comparisons with *SMiRL Finetuning*, where the stable skills for humanoid keep the average height around 1. (right): Training process. The decrease in the variance of the height implies the emergence of a stable skill.

**Behaviors in stable environments.** For the second question, we apply our method to state-reaching tasks: $g \triangleq \text{Relabel}(\pi_\mu, \omega, \tilde{\tau}) = \tilde{s}_T$ (Figure 3 c2). We adopt the negative L2 norm (between the goal and the final state in each episode) as the distance metric. We compare our method (*DARS*) against six alternative goal-reaching strategies[4]: (1) additionally updating $\pi_\theta$ with data $\mathcal{B}_\mathcal{T}$ collected in the target (*DARS Reuse*); (2) employing DARC with a negative L2-norm-based reward (DARC L2); training skills with GPIM in the source and target respectively (3) *GPIM in source* and (4) *GPIM in target*); (5) updating GPIM in the target 10 times more (*GPIM in target X10*; $R = 10$ and see more interpretation in [11]); (6) finetuning *GPIM in source* in the target (*GPIM Finetuning in target*).

We report the results in Figure 8. *GPIM in source* performs much worse than *DARS* due to the dynamics shifts as we show in Section 5.1. With the same amount of rollout steps in the target, *DARS* achieves better performance than *GPIM in target X10* and *GPIM Finetuning in target*, and approximates *GPIM in target* within 1M steps in effectiveness, suggesting that the modification $\beta\Delta r$ provides sufficient information regarding the target dynamics. Further, reusing the buffer $\mathcal{B}_\mathcal{T}$ (*DARS Resue*) does not significantly improve the performance. Despite not requiring a prior reward function, our unsupervised DARS reaches comparable performance to (supervised) *DARC L2* in *(Map-a, Map-b)* pair. The more exploratory task *(Map-b, Map-c)* further reinforces the advantage of our dynamics-aware rewards, where the probing policy $\pi_\mu$ boosts the representational potential of $q_\phi$.

**Behaviors in unstable environments.** Further, when we set $p(\omega)$ as the Dirac distribution, $p(\omega) = \delta(\omega)$, the discriminator $q_\phi$ will degrade to a density estimator: $q_\phi(s_{t+1})$, which keeps the same form as in SMiRL [4]. Assuming the environment will pose unexpected events to the agent, SMiRL seeks out stable and repeatable situations that counteract the environment's prevailing sources of entropy.

With such properties, we evaluate DARS in unstable environment pairs, where the source and the target are both unstable and exhibit dynamics mismatch. Figure 9 (left) charts the emergence of a

---

[4]We do not compare with other unsupervised RL methods (eg. Warde-Farley et al. [53]) because they generally study the rewards wrt the high-dimensional states. DARS does not focus on high-dimensional states. Domain randomization [35, 50] and system (dynamics) identification [12, 7, 1] are also not compared because they requires the access of the physical parameters of source environment, while we do not assume this access.

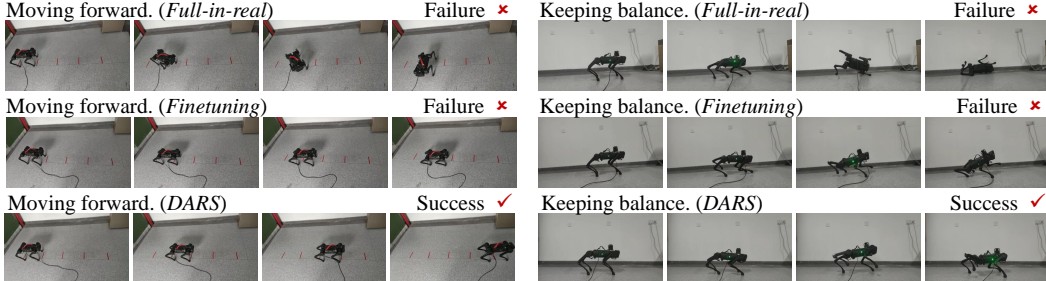

Figure 10: Deploying the learned skills into the real quadruped robot, where all models are trained with limited interaction (three hours for moving forward and one hour for keeping balance) in real.

stable skill with DARS, while SMiRL suffers from the failure of domain adaptation for both *(H, A-H)* and *(H, B-H)*. Figure 9 (right) shows the comparisons with *SMiRL Finetuning*, denoting training in the source and then finetuning in the target with SMiRL. With the same amount of rollout steps, we can find that *DARS* can learn a more stable skill for the target than *SMiRL Finetuning*, revealing the competence of our regularization term for learning adaptive skills even in the unstable environments.

### 5.3 Sim2real Transfer on Quadruped Robot

We now deploy our DARS on pair *(sim-robot, real-robot)* to learn diverse skills (moving forward and moving backward) and balance-keeping skill in stable and unstable setting respectively. We compare DARS with two baselines: (1) training directly in the real world (*Full-in-real*), (2) finetuning the model, pre-trained in simulator, in real (*Finetuning*). As shown in Figure 10, after three hours (or one hour) of real-world interaction, our DARS demonstrates the emergence of moving skills (or the balance-keeping skill), while baselines are unable to do so. As shown in Table 1, *Fine-*

Table 1: Time (hours) spent for valid skill emergence in real-world interaction (covering the manual reset time).

|  | forward & backward | keeping balance |
|---|---|---|
| Full-in-real | > 6 h | > 6 h |
| Finetuning | > 6 h | 4 h |
| DARS | 3 h | 1 h |

*tuning* takes significantly more time (four hours vs. one hour) to discover balance-keeping skill in the unstable setting, and the other three comparisons are unable to acquire valid skills given six hours of interaction in the real world. Supplementary material contains videos from this sim2real deployment.

## 6 Conclusion

In this paper, we propose DARS to acquire adaptive skills for a target environment by training mostly in a source environment especially in the presence of dynamics shifts. Specifically, we employ a latent-conditioned policy rollouting in the source environment to represent goals (including goal-distribution and goal-achievement reward function) and introduce a KL regularization to further identify consistent behaviors for the goal-conditioned policy in both source and target environments. We show that DARS obtains a near-optimal policy for target, as long as a mild assumption is met. We also conduct extensive experiments to show the effectiveness of our approach: (1) DARS can acquire dynamics-aware rewards, which further enables adaptive skills for the target environment, (2) the rollout steps in the target environment can be significantly reduced while adaptive skills are preserved.

## Acknowledgments and Disclosure of Funding

The authors would like to thank Hongyin Zhang for help with running experiments on the quadruped robot. This work is supported by NSFC General Program (62176215).

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
