# OpenReview forum: "Unsupervised Domain Adaptation with Dynamics-Aware Rewards in Reinforcement Learning"
_NeurIPS.cc/2021/Conference — NeurIPS 2021 Poster_

### Official Review · Reviewer_3aSe · 2021-07-10

**Rating:** 7
**Confidence:** 4

**Summary:**

The paper handles the problem of domain adaptation in RL, specifically the setting where rewards and prior goal representations are not available. They do so by combining a skill learning method (in the source environment) with a transfer learning method that aims to match the joint $(goal, \tau)$ distribution arcoss environments. By doing so they show that skills can both be learned in the source domain, and they transfer more easily to the target domain.

**Limitations And Societal Impact:**

Yes

**Main Review:**

The authors want to use domain adaptation to learn a policy that will perform well in a target environment, assuming access to a source environment where it is easier to get many samples. In addition, the authors want a method that will do so without access to rewards in either the source or target environment.

They propose a method called DARS, standing for dynamics-aware rewards. It is based off 2 primary components. First, there is a skill learning policy. For this policy, they use a common trick in the literature: an exploratory policy $\pi$ is conditioned on some latent skill vector $\omega$. As trajectories are generated, a learned $q_\phi(\omega|s)$ acts as a discriminator that tries to identify $\omega$ from the episode. Intuitively, if the mutual information between $\omega$ and episode $\tau$ is maximized, then different $\omega$ should lead to distinguishable trajectories in the environment. (The specific citations for this are DIAYN and GPIM but something analogous to this has shown up in other recent papers as well.)

They then combine this with a dynamics based reward. The aim is to match distributions $(goal, \tau)$ across the source environment and target environment, given the same policy $\pi$. Through some derivation, you can show that when taking the KL divergence $p(g,\tau)$, all terms between the two cancel out except for $p_{source}(s_{t+1}|s_t,a_t)$ and $p_{target}(s_{t+1}|s_t,a_t)$. The ratio of these probabilities can then further be approximated by learning 2 classifiers $p(target|s_t,a_t,s_{t+1})$ and $p(target|s_t,a_t)$. This closely follows the Domain Adaptation with Rewards from Classifiers (DARC) paper.

Combined, they demonstrate that skills learned in the source domain transfer better to the target domain.

I found the presentation in this paper a bit dense, although that is to be expected since they are aiming to explain 2 papers worth of ideas in 1. At a surface level, I personally found it confusing that $\Delta r$ was defined opposite to how it was defined in the DARC paper (i.e as (source - target)), and I initially thought this was an error until I noticed that it was subtracted instead of added to the reward. I think negating it and treating $\Delta r$ as an additive reward bonus is slightly more intuitive.

This is a fairly minor issue, my more detailed reservation with the paper is that it doesn't feel like a very interesting research contribution. To me, it feels like it is primarily about taking DARC then replacing the source rewards with a skill learning method. The 2 working together doesn't seem novel to me because they are somewhat orthogonal methods (one is skill learning in 1 env and the other is transfer learning between 2 envs). Both are based on defining rewards for an RL method to optimize. A priori I would assume the two would work together as long as you balanced their reward objectives appropriately, and this is basically what the paper shows (adjusting the $\beta$ weight is what affects whether the method works or not). Moreover it doesn't seem like their comparisons are very strong - they are essentially comparing against a skill learning / intrinstic motivation (SMiRL, GPIM), or they are comparing against a transfer learning method (DARC with a proxy reward), and again it seems like the method is supposed to work if tuned properly. And it does, by a margin that appears small.

To me, if it a priori is supposed to work, and the improvement is fairly small, then I'm not sure the paper needs to exist. This is especially true because the target problem they are solving feels narrow, as it is the intersection of 2 other problems. They aim to solve "transfer learning, between 2 environments, where we have ready access to 1 and not the other, but we also don't have rewards in the source env we have ready access to", and this doesn't feel like a common scenario of interest, which weakens significance.

Without an especially strong empirical result, to be strong enough for acceptance the paper needs some argument that the combination of the two is better than the sum of its parts. It attempts to argue this by saying that the dynamics from the target domain are affecting the skill learning process (in the Map env visualizations). I think this is basically just a consequence of "weighting the objectives correctly" - in the limit, at $\beta = 0$ you recover DIAYN and at $\beta = \infty$ you recover DARC, so at some $\beta$ in between you expect to see episodes that 1) are closer to the target episodes but 2) still have diversity within that restricted part of state space, and that's effectively what we see here.

Edit: I have read the other reviews and rebuttals. Please see the comment later in the thread for more detailed replies. In short,

* I agree that $\beta$ does not inherently limit diversity of skills, it is primarily about alignment in the target domain. This may reduce diversity of states that are reached in the source domain, but this is fine because the end goal is to learn diverse skills for the target domain - anything that is closed off by the $KL$ objective is something we should not learn anyways.
* One of my comparisons was comparing an unsupervised (no observed reward) method against a supervised (observed reward) method, which was a mistake on my part.
* After some more consideration, I believe the problem the paper aims to solve is more important than I previously thought.

These were significant misunderstandings so I am updating from 4 to 7.

**Time Spent Reviewing:**

2hr 15min, additional 3 hours post rebuttal.

---

> ### Author Response · Authors · 2021-08-10
> **Thank you for your questions! Please see our response below (WITH NEW SUMMARY)**
>
> We thank the reviewer for the work, but given the review we are afraid the reviewer may have misunderstood the point of our paper and did not provide a fair assessment of our contribution.
>
> We hope our responses below and the comments of the other reviewers may help clarify the scope of our work and its significance.
>
> **(1) "Weighting the objectives."**
>
> We respectfully disagree with the reviewer that DARS is based on two components --- *[one skill learning term]* and *[one KL divergence term]* (Section Main Review, the second and third paragraphs). Note that we introduce *[the skill learning term]* in Equation 4, not in Equation 3. Following the reviewer's comments, we can find that the two components in Equation 4 are *[one skill learning term] $I(\omega; \tilde{\tau})$* and *[the difference term between mutual information and KL] $I(g;\tau) - \beta KL$*, which is conflict with the two components mentioned by the reviewer.
>
> We think the reviewer might have overlooked the relationship between Equations 3 and 4. We claim that Equation 3 refers to the objective of unsupervised domain adaptation, and Equation 4 refers to the objective of our DARS. In Equation 4, we can see that DARS dose not explicitly weight *[the skill learning term]* against *[the difference term between mutual information and KL]*, where the coefficients for the two terms are all equal to one. The reviewer takes $\beta$ as such a coefficient and mistakenly asserts that $\beta$ affects the diversity of induced states (Section Main Review, the last line of the last paragraph).
>
>
>
> **(2) The role of $\beta$.**
>
> Another misunderstanding mentioned by the reviewer is that "at $\beta=\infty$ DARS recover DARC". In fact, at $\beta=1$ instead of $\infty$, we recover (unsupervised) DARC.  And we emphasize that $\beta$ is not such a coefficient that balances between skill learning objective and DARC-style objective.
>
> By introducing $\beta$ in Equation 3, we have changed the objective of vanilla DARC. And this change provides a principled method for increasing or decreasing the weight on the $\Delta r$ term, depending on how much the policy is currently exploiting the dynamics discrepancy.
>
> ***A short summary of answers (1) and (2)*:** our objective is not simply "weighting the objectives correctly". 1) we do not explicitly weigh the two objectives (skill learning and DARC-style objectives); 2) $\beta$ balances $q_\phi$ and $\Delta r$ for the goal conditioned policy $\pi_\theta$ and does not affect the probing policy $\pi_\mu$.
>
>
>
> **(3) Combining skill learning (eg. DIAYN) and DARC: "It feels like it is primarily about taking DARC then replacing the source rewards with a skill learning method".**
>
> The reviewer mentions that "skill learning and DARC are based on defining rewards for an RL agent to optimize" and claims that "they are orthogonal". We show in Figure 1 that skill learning is more than defining rewards $r_g$, but concurrently finding the goal distribution $p(g)$. That is, we need to capture the interplay between skill learning and DARC-style objectives, so as to make both $r_g$ and $p(g)$ to be adaptive for the target domain, rather than to combine DIAYN and DARC in isolation from two orthogonal problem settings.
>
> Figure 1 (right) shows that *both source and target environments* shape $r_g$ and $p(g)$. In other words, skill learning provides $r_g$ and $p(g)$ for the DARC-style objective, and in turn, DARC-style objective impacts the skill learning process. We focus on this interplay setting, instead of building upon DARC with learned rewards. In our supplementary material, we further verify such interplay, where dynamics shifts change the goal distribution for source and target. Directly combining skill learning (eg. DIAYN) and DARC fails to learn an adaptive policy conditioned on shifted goals, while a simple extension can be derived from our framework to relax such shifts.
>
> In this view, our method expands the scope and utility of adaptive policy, which scales much better than the primary about "taking DARC then replacing the source rewards with a skill learning method".
>
>
>
> **(4) Performance improvement is "small".**
>
> We first argue that the improvement is not small. We compare DARS against (unsupervised) skill learning method and (supervised) DARC in both stable and unstable environments, showing that  1) vanilla skill learning method fails to acquire an adaptive policy for the target environment;  2) (unsupervised) DARS acquires comparable performance with (supervised) DARC, and can even surpass (supervised) DARC in some tasks (eg. Map-a ---> Map-b);  3) DARS with stronger $\beta$ ($\beta>1$) achieves better performance than DIAYN+DARC ($\beta=1$).
>
> We also deploy our DARS on the sim2real setting where online data (in real) collection is quite expensive. The benefits of our DARS enable us to acquire locomotion skills without any rewards and with limited interaction in the target environment. To our knowledge, DARS is the second successful work learning skills in real robot in unsupervised setting. The first work is off-DADS [1], where the emergence of skills takes about 20 hours, while DARS acquires skills only under 3 hours. (Note that we adopt different robots, thus this comparison is not absolutely  fair.) Moreover, in the offline setting (for the expensive target environment), we further reduced the time to 2.25 hours. See Question (1) of Reviewer UYK4.
>
> [1] Emergent Real-World Robotic Skills via Unsupervised Reinforcement Learning, Sharma et al.
>
>
>
> We hope this addresses the reviewer’s concerns and welcome any further feedback.
>
>
>
> ---
> ---
>
> NEW:
>
> **A brief summary of the previous points:**
>
> |                       | Unsupervised | Domain Adaptation | Target-Dynamics-Oriented Reward $r_g$ | Target-Dynamics-Oriented Goal Distribution $p(g)$ |
> | :-------------------: | :----------: | :---------------: | :-----------------------------------: | :-----------------------------------------------: |
> |    Skill Learning     |   $\surd$    |         x         |                   x                   |                         x                         |
> |         DARC          |      x       |      $\surd$      |                $\surd$                |                         x                         |
> | Skill Learning + DARC |   $\surd$    |      $\surd$      |                $\surd$                |                         x                         |
> |      DARS (ours)      |   $\surd$    |      $\surd$      |                $\surd$                |                      $\surd$                      |
>
> 1. Motivation and problem setting
>
>    Our motivation is not the same as skill learning and DARC, but an extension of them. Our problem is common in real world. For example, the real robot is required to acquire skills while it is expensive to design the goal representations ($r_g$ and $p(g)$) and run $\pi_\theta$ in real so as to train the policy $\pi_\theta$.
>
> 2. "Combining skill learning and DARC"
>
>    We are not just combining skill learning and DARC. The simple combination of skill learning and DARC presents practical challenges (eg. facing goal distribution shift between source and target environments).
>
> 3. Experiment
>
>    Our experiments give new insight, showing that our learned goal representations ($r_g$ and $p(g)$) are well shaped by both dynamics.

---

> > ### Comment · Reviewer_3aSe · 2021-08-20
> > **Re: author rebuttal**
> >
> > I thank the authors for their reply - comments for specific sections follow
> >
> > **(1) + (2) "Weighing the objectives" and the role of $\beta$**
> >
> > Looking more closely at the objective, if $\beta = \infty$, there can be several solutions where $KL(p_{p_S,\pi}(g,\tau) || p_{p_T,\pi}(g,\tau)) = 0$, and the overall objective will still encourage maximizing mutual information. My misunderstanding was that I assumed $\beta = \infty$ would only allow for 1 solution, but in reality $\beta = \infty$ would just change the problem to a constrained optimization problem, and does not inherently limit the skills that could be learned.
> >
> > I still believe that $\beta$ does affect the probing policy $\pi_\mu$ in an indirect way, in that it affects the KL loss, which depends on goals $g$, which are either directly $\omega$ or relabeled from data generated by $\pi_\mu(a|s,\omega)$, but this resolves my concerns and I agree that $\beta$ does not necessarily limit diversity.
> >
> > **(3) Combining skill learning and DARC**
> >
> > I recognize that the target environment dynamics have an impact on the learned skills. My disagreement is on how important this is. Here is my understanding of the baselines:
> >
> > * DARS
> > * DARS where it is allowed to update using target env data (DARS reuse)
> > * DARC with a reward based on L2 norm between states (DARC L2)
> > * GPIM, either learned directly in source or target, or learned in source then finetuned in target.
> >
> > To be convinced of the importance of interplay between target environment dynamics, and skill learning in source, I would want to see results where a skill learning policy is learned entirely in the source environment, followed by a pure transfer learning stage where a method like DARC (or something similar) is used to adapt the source skill policy into the target environment. This would directly examine whether there was benefit to incorporating target environment information during the skill learning process, and it was unclear to me whether the baselines did such a comparison.
> >
> > It is possible that "GPIM, finetuned in target" is such a baseline, but if it is, then this could be explained more in the paper. The details of this baseline were fairly sparse and I did not find an elaboration in the appendix - therefore I did not consider it when writing my review.
> >
> > **(4) Performance increase is small**
> >
> > I was comparing DARS against the supervised DARC L2 curve when making this comment - I apologize for missing this detail and agree the performance is significant over other unsupervised baselines.
> >
> > **(5) Applicability of method**
> >
> > The core disagreement of my review was around how common the research problem was. Often, papers construct best-case evaluation scenarios that best empahsize settings their method performs well in, but what matters for the research is how often such scenarios arise in practice.
> >
> > My impression when first reading the paper was that the setting would be fairly uncommon. Normally, it is very odd to both have easy access to a source env (like simulation), and also not have access to a reward function. Even in settings where it is hard to define an appropriate reward, it is often possible to define a reward function from data (which is plentiful).
> >
> > My belief was that the authors viewed their work as learning for the purpose of learning downstream tasks (forward & backward, keeping balance) more quickly. And therefore, it seemed very odd that there were minimal attempts to learning those downstream tasks directly.
> >
> > However, based on the author rebuttal, I see that the authors believe the main contribution of their work is skill learning in the target environment, rather than specific downstream tasks of interest. In particular, the behaviors in Table 1 of Section 4.3 are not the direct tasks the authors care about, they are instead skills that emerge out of DARS, and the claim is about the amount of time needed for such skills to emerge.
> >
> > So, the question becomes this: how often do we want to apply a skill learning method on a real robot? After thinking about it, I agree this is a problem of greater significance.
> >
> > For those reasons I plan to increase my rating.

---

> > > ### Author Response · Authors · 2021-08-24
> > > **We thank the reviewer for detailed comments**
> > >
> > > Thank you! We are very glad that you agree with the importance of our problem, and we appreciate your insightful comments.
> > >
> > > (3) "It is possible that "GPIM, finetuned in target" is such a baseline (Q 3)"
> > >
> > > Yes, you are right. We will add more details to the experiment (baselines).
> > >
> > >
> > > Thanks again for taking the time to reply.

---

### Official Review · Reviewer_kXzn · 2021-07-15

**Rating:** 7
**Confidence:** 3

**Summary:**

The authors address the problem of unsupervised Domain Adaptation in Reinforcement Learning. Combining ideas from previous work, they devise a method able to utilize an easy to sample source environment (e.g., a simulator) in order to learn a set of skills without supervision, while at the same time guarantee that these skills will be suited to the target environment (e.g., a real robotic system). To achieve this, they modify the reward structure of the typical Unsupervised Reinforcement Learning setup to enable the agent to build skills that are compatible with the target environment dynamics, while penalizing exploration in regions of the source environment where the dynamics differ.
The proposed approach is benchmarked against state-of-the-art algorithms in a wide variety of environments with increased complexity and manages to outperform previous baselines both in simulation as well as in three walking tasks on a real quadruped robot.


**Limitations And Societal Impact:**

The first part of the paper starts with a very clear and well-written introduction in Sections 1 and 2 that introduces the problem and the proposed approach. Figures 1 and 2 provide an excellent overview and help understand the intuition behind the setup and the proposed algorithm. This holds for Sections 3.1 and 3.2 (aided by the visual explanation in Figure 3), where the problem is described in detail along with the derivation of the modified objective in Equation 4.
Unfortunately, Sections 3.3 – 3.5 which describe the details of the proposed approach are difficult to follow, as the authors omit critical information. The definition and purpose of the discriminator q_\phi is never explained (or discussed). The same goes for the two classifiers q_\psi: they are introduced directly in Algorithm 1 and referenced in one line in Section 3.5. Also, it is not clear by reading the text how they are used to shape the reward \Delta r. If the reader follows the references and reads the relevant papers, these design choices become clear, but the paper should be made more self-contained.
I would recommend the authors to bring Figure 10 of Section F.1. of the supplementary material in the main text and use it to explain what is the role of the discriminator, how the classifiers are trained and how they are used to calculated \Delta r.
The experiments Section is quite dense and difficult to follow at some points. For example, I would recommend moving the comparisons shown in Figures 8 and 9 in the appendix and expand the discussion there a bit.

Pros:
- Very well written for the most part (see discussion on problematic Sections above)
- Figures providing intuition on the proposed method, as well as visualization of skills (both trajectories and heatmaps) are very well thought and complement the text in an excellent way
- Very extensive experimental evaluation and impressive results on real robot

Limitations:
- One of the main limitations is that the approach requires full knowledge and a-priori access to the target environment, while at the same time assuming that the goal distribution is the same between the two. Nevertheless, the authors discuss these limitations in Appendices B and D providing insights and extensions to the algorithm.


**Main Review:**

The authors address the problem of unsupervised Domain Adaptation in Reinforcement Learning. Combining ideas from previous work, they devise a method able to utilize an easy to sample source environment (e.g., a simulator) in order to learn a set of skills without supervision, while at the same time guarantee that these skills will be suited to the target environment (e.g., a real robotic system). To achieve this, they modify the reward structure of the typical Unsupervised Reinforcement Learning setup to enable the agent to build skills that are compatible with the target environment dynamics, while penalizing exploration in regions of the source environment where the dynamics differ.
The proposed approach is benchmarked against state-of-the-art algorithms in a wide variety of environments with increased complexity and manages to outperform previous baselines both in simulation as well as in three walking tasks on a real quadruped robot.


**Time Spent Reviewing:**

10

---

> ### Author Response · Authors · 2021-08-10
> **Thank you for the comments and suggestions! Please see our response below**
>
> We’d like to thank the reviewer for their thorough review and helpful suggestions.
>
> As suggested by the reviewer, we will add
>
> + the definition and purpose of the discriminator $q_\phi$,
> + the training details of the two classifiers $q_\phi$ wrt. $\Delta r$ and
> + more analysis about the experiments.
>
> **(1) "The approach requires full knowledge and a-priori access to the target environment."**
>
> To relax such access to the target environment (online interaction), we think a promising direction is considering the offline setting for the target environment.  We can directly extend our DARS on this setting and reuse the unlabeled experience, data of mixed and unknown quality without reward annotations, to provide the modification term $\Delta r$ for dynamics shifts. Moreover, our preliminary experiments under such offline setting (for the target environment) have shown that adaptive skills emerge in stable environments.  Please refer to Question (1) of Reviewer UYK4.

---

> > ### Comment · Reviewer_kXzn · 2021-08-24
> > **Re. author rebuttal**
> >
> > Thank you for your response and thank you for providing the pointer concerning the a-priori access to the target environment.

---

### Official Review · Reviewer_UYK4 · 2021-07-21

**Rating:** 8
**Confidence:** 3

**Summary:**

This paper presents an approach for unsupervised learning of a goal-conditioned policy that is robust to changes in dynamics and can be transferred to a target domain that is different from the source domain in terms of the underlying dynamics. There are several components. First, a latent-conditioned probing policy is used to generate goals (either in the source or target domain). This probing policy is trained to generate diverse goals by maximising the information gain between the goals and the resulting trajectories generated by the policy, i.e. we want diverse goals that are predictable given the resulting trajectories. This “predictability” is approximated using a discriminator that scores the likelihood of the goal latent given the resulting trajectory; this is also trained alongside the goal-generating policy based on rollouts in the source domain.

Next, the probing policy is used to generate goals in the source and target domains (these are either the latents themselves or the final states of trajectories from the probing policy), and the goal-conditioned policy (which is conditioned on these goals) is executed in both these domains to generate rollouts. A key assumption here is that it is costly to execute the policy in the target domain (e.g. real world), so the approach tries to minimise this by training the probing policy, discriminator and to a major extent, the policy, primarily on the source domain. The policy is again trained to maximise the log-likelihood of the discriminator, thereby leading to predictable trajectories in the source domain. To ensure that the policy does not overfit to the source domain, and consequently fail on the target domain, an additional KL regularisation term is added to the policy training objective. This regularisation penalises trajectories that provide inconsistent results between the source and target domains, as measured by the KL divergence of the resulting transition distributions when executing the policy for the same goal in both these domains. This effectively ensures that the policy learns trajectories that are executable consistently in both domains, thereby ensuring success even in the presence of differences in dynamics between the two domains. Note that two key assumptions that are necessary for this to work (and the theoretical results) is that there is no transition in the target domain that is not possible in the source domain, and that the goal distribution in the source is also valid in the target (a relaxation of the latter is provided in the supplementary material). This regularisation term is approximated by a pair of classifiers that are trained on data from both domains; these are also trained alongside the policy. The final algorithm alternates training the probing policy & discriminator that parameterise the goal distribution and rewards against training the goal-conditioned policy and the classifiers for approximating the dynamics difference KL.

The approach is tested on several simulated tasks of varying complexity. First, the approach is tested on a 2D point mass with pairs of source and target tasks where the arena has walls of different configurations. Next, the approach is evaluated on continuous control tasks from the OpenAI Gym with the ant, half-cheetah and humanoid embodiments. The source tasks are the standard gym tasks and the target tasks have certain joints not working or additional perturbations in the form of obstacles. Lastly, the approach is also tested in a sim2real setup with a quadruped; the source task is in simulation while the target task is on the real robot. The resulting policy and goal distribution show good performance on all these tasks; the skills and goal distribution learned by the method is able to successfully transfer across changes in wall configuration, joint issues, stability problems and also to real world robots. Several visualisations of different parts of the algorithm (e.g. discriminator, skills) etc. are provided along with comparisons to baselines, namely variants of DARC, SMiRL, DIAYN and GPIM. The supplement also provides additional analysis and proofs of the theoretical guarantees of the near-optimality of the proposed method under the pertinent assumptions.

**Limitations And Societal Impact:**

The authors have clearly explained the limitations of the approach, explaining in detail the assumptions that are needed to get the approach working (and for the theoretical results), the resulting restrictions on the method and potential relaxations for some of the assumptions with experimental results for these relaxations. Brief discussions of the broader impact is provided in the supplement, though some more details could be provided with regards to the social impact (especially since the approach is applicable for sim2real).

**Main Review:**

This paper presents an approach to unsupervised RL that trains a goal-conditioned policy which is capable of adapting to changes in dynamics between a source and target domain. A key advantage of the method is that it is able to train in an unsupervised fashion in target domains where data collection can be hard or expensive. This is done by training the goal distribution, goal-based reward function and goal-conditioned policy primarily on the source domain where data collection is inexpensive, while regularising for differences in dynamics between the two domains. This regularisation encourages the policy to find skills that lead to consistent results in the source and target domains while utilising an order of magnitude lower amount of data from the target domain. This is particularly promising for real world robotics, especially in the context of sim2real methods, where simulated data is easy and cheap to generate but has differences in dynamics to the real world where data is also expensive to generate.

The approach is tested on several simulated and real world source and transfer tasks and shows promising results. There are several assumptions that are key for the method to work. Key amongst those is the assumption that there is no transition that is possible in the target domain but impossible in the source domain; interestingly, one of the point-mass experiments does violate this assumption and while the approach does work on this transfer setting it effectively over-constrains the search space leading to a potentially sub-optimal solution. It would be interesting to see if this assumption can be relaxed. Another important assumption is that the goal distribution is consistent under dynamics changes between the source and target domains; an approach to relax this assumption is derived in the supplementary material wherein the goal distribution training can be regularised to respect the source/target dynamics changes and is validated with a set of experiments (it would be useful in this context to see (Map-a, Map-e) discriminator visualisations in Figure 6).

Overall, I am quite encouraged by the results presented in the paper, which are strong and show that the method is able to do well on a variety of control tasks both in simulation and in the real world. I am particularly excited about the applications to sim2real settings where the data limitations can make learning quite hard. It would be interesting to see if further reductions in the data is possible, and it would be great if there was an ablation study with respect to the number of real world data points used (i.e. fraction of target domain data) on some of these tasks. To add further clarity to the paper it would be useful if the approach for training the classifiers to estimate the dynamics regularisation is also detailed either in the main text or supplementary material.

**Time Spent Reviewing:**

5 hours

---

> ### Author Response · Authors · 2021-08-10
> **Thank you for the comments and suggestions! Please see our response below**
>
> Thank you for the great summarization of our main contributions and we really appreciate your encouraging comments.
>
> **(1) "It would be interesting to see if further reductions in the data is possible, and it would be great if there was an ablation study with respect to the number of real world data points used (i.e. fraction of target domain data) on some of these tasks."**
>
> Yes, that is a great point. We consider two settings to see if further reductions in the data (for the expensive target environment) are possible.
>
> 1) Online setting. Figure 8 shows the performance as we run policy $\pi_\theta$ in the target environment. To reduce the online steps in the target, we investigate whether larger $R$ (more steps in the source environment) can provide richer training signals as we train $\pi\theta$ in source (which is cheap). We pick the (Map-a, Map-b) task to validate this supposition. We show our results in https://sites.google.com/view/Anonymous-dars, finding that a large $R$ is not sufficient to reduce the online rollout steps in the target environment.
>
> 2) Offline setting. To reduce the time required for collecting data in the target environment, we can instead allow our method to use previously collected data to aid the training. Specifically, we only need to replace steps 13-16 in Algorithm 1 with prior offline data. For (Map-a, Map-b) and (Map-b, Map-c) tasks, we adopt another random (behavior) policy to collect data for the target environment. Then we show the performance (distance to goals vs. steps of the random behavior policy in the target) in the following table.
>
> |                  | 0k                | 10k               | 20k               | 30k               | 40k               | 50k               |
> | ---------------- | ----------------- | ----------------- | ----------------- | ----------------- | ----------------- | ----------------- |
> | Map-a ---> Map-b | 0.224 (+/- 0.047) | 0.069 (+/- 0.048) | 0.057 (+/- 0.041) | 0.055 (+/- 0.037) | 0.043 (+/- 0.033) | 0.039 (+/- 0.030) |
> | Map-b ---> Map-c | 0.381(+/- 0.026)  | 0.115 (+/- 0.066) | 0.097 (+/- 0.057) | 0.084 (+/- 0.064) | 0.082 (+/- 0.053) | 0.078 (+/- 0.050) |
>
> Compared with the results in Figure 8, such offline training does reduce the rollout steps in the target environment while maintaining comparable performance. In our anonymous site, we also show the results for half-cheetah and ant tasks under the offline setting.
>
> Moreover, we demonstrate applying the offline setting to our sim2real experiment. To obtain the offline data for the real robot, we reuse previous data collected from running $\pi_\theta$ in real. Then we test whether training on less data leads to our adaptive policy. We divide the data (3 hours for forward and backward tasks, Table 1)  into four equal parts. We use 1111 to denote the complete data, and replace 1 with 0 to denote discarding the corresponding offline data fraction. For example, 0111 refers to the last three quarters of data, and 1100 refers to the first half of the data. We report our experiment in the table below.
>
> | 2.25 hours offline data | 0111    | 1011    | 1101    | 1110    |
> | ----------------------- | ------- | ------- | ------- | ------- |
> | Forward                 | success | success | success | success |
> | Backward                | success | success | success | failure |
>
> | 1.5 hours offline data | 0011    | 0101    | 0110    | 1001    | 1010    | 1100    |
> | ---------------------- | ------- | ------- | ------- | ------- | ------- | ------- |
> | Forward                | success | success | failure | failure | failure | failure |
> | Backward               | failure | failure | failure | failure | failure | failure |
>
> See videos at https://sites.google.com/view/Anonymous-dars. This experiment shows that we could reduce the number of real world data points used while keeping comparable performance. Note that in the offline sim2real setting, we do not carry out careful tuning (eg. the force and frequency of the real robot), with which we think better performance can be achieved. We will further study the performance of other skills on the real robot under the offline setting, seeking progress towards the conveying larger point of reward-free learning being realized in real-world.
>
>
> **(2) "It would be useful if the approach for training the classifiers to estimate the dynamics regularization is also detailed either in the main text or supplementary material."**
>
> We will add the training details of the classifiers in the supplementary material.
>
> **(3) "More details could be provided with regards to the social impact (especially since the approach is applicable for sim2real)."**
>
> Thank you for the suggestion. We will add more social impacts (challenges, progress, and applications) with respect to the sim2real setting.

---

### Decision · Program_Chairs · 2021-09-27

**Decision:**

Accept (Poster)

**Comment:**

The reviewers highly appreciated the author response and the additional details. Reviewer UYK4 didn't update the review but indicated in the discussion to have read the replies.
We had quite an extensive discussion between the reviewers on whether the setting of this paper too narrow and whether that would actually speak against publishing it. In the end they came up with a few examples (please add some motivating examples that have practical relevance to the paper) and everybody was convinced that this is a sound and interesting paper.